# Comprehensive Map of Canine Angiostrongylosis in Dogs in Spain

**DOI:** 10.3390/ani12172217

**Published:** 2022-08-29

**Authors:** Elena Carretón, Rodrigo Morchón, Sara Nieves García-Rodríguez, Iván Rodríguez-Escolar, Jorge Isidoro Matos, Noelia Costa-Rodríguez, José Alberto Montoya-Alonso

**Affiliations:** 1Veterinary Medicine and Therapeutic Research, Faculty of Veterinary Medicine, Research Institute of Biomedical and Health Sciences (IUIBS), University of Las Palmas de Gran Canaria, 35413 Arucas, Spain; 2Zoonotic Diseases and One Health Group, Laboratory of Parasitology, Faculty of Pharmacy, University of Salamanca, Campus Miguel Unamuno s/n, 37007 Salamanca, Spain

**Keywords:** *Angiostrongylus vasorum*, Spain, Europe, epidemiology, dogs, provinces, autonomous communities, animal diseases

## Abstract

**Simple Summary:**

*Angiostrongylus vasorum* (*A. vasorum*) is the causal agent of canine angiostrongylosis, an emerging disease in Europe. In Spain, *A. vasorum* mainly affects wild animals (red foxes and wolves), although studies in domestic dogs are scarce but show evidence of an expansion of the disease. The aim of this study was to analyze the presence of *A. vasorum* in domestic dogs throughout Spain, taking into account that there were still provinces where this canine infection has not been studied. Blood samples from 5544 domestic dogs were collected from January 2020 to March 2022. These samples were tested for the presence of circulating *A. vasorum* antigens. The overall prevalence of canine angiostrongylosis in Spain was 1.39%. No significant differences were found for sex and age, but significant differences were found for habitat. Infected domestic dogs were reported in most Spanish provinces, with lower prevalence observed for inland provinces and higher prevalence observed for provinces along the coast, where climatic factors seem to be determinant in the establishment of the parasite.

**Abstract:**

Canine angiostrongylosis is an emerging disease caused by *Angiostrongylus vasorum*, mainly affecting wild carnivores and dogs. In Spain, there are studies reporting infections in foxes, wolves, and badgers in different regions of the country. However, there are hardly any publications on its prevalence in dogs. The aim of this study was to complete and update the epidemiologic map of *A. vasorum* in dogs in Spain. A total of 5619 canine blood samples from all autonomous cities and provinces of Spain were collected and tested for the presence of circulating *A. vasorum* antigens. The overall apparent prevalence of canine *A. vasorum* infection in Spain was 1.39%. No significant differences were found for sex or age, but significant differences between outdoor and indoor/outdoor dogs were found. A high prevalence was also observed in the northern third of the country, where an oceanic climate prevails, being humid and rainy and where abundant vegetation can be found, thus favoring the proliferation of intermediate hosts. The results suggest that *A. vasorum* canine infections are heterogeneously present in a large part of the territory, demonstrating its expansion throughout the country, and therefore, awareness and prevention campaigns for this disease should be promoted.

## 1. Introduction

Canine angiostrongylosis is a disease caused by the parasitic nematode *Angiostrongylus vasorum* (Baillet, 1866), which mainly affects canids, both domestic and wild, with foxes, jackals, wolves and domestic dogs being the most studied hosts [1,2]. The life cycle of the parasite is indirect, so it involves intermediate hosts, usually snails or slugs, or paratenic hosts (frogs and domestic chicken), which carry the infective larval L3 form [3,4]. The adult forms of the parasite inhabit the pulmonary arteries and the right ventricle of the heart of the definitive host, while after ingestion, the L3 migrates from the intestines to the mesenteric lymph nodes and, subsequently, after a molting period, to the right ventricle through mesenteric lymphatic vessels, the portal vein, hepatic veins, and the caudal vena cava [5,6,7]. Adult females produce eggs that embryonate and hatch within the pulmonary capillaries, where L1 penetrates the alveoli to subsequently migrate through the trachea and larynx to the oral cavity. Once there, they are swallowed, travel throughout the digestive system, and are eliminated through feces. In the environment, L1 penetrates the intermediate host and molt to the infective stage L3.

This parasite can cause a wide range of clinical signs in the infected animal, including hemorrhagic (bleeding diathesis) and neurological signs. However, the most frequent clinical presentation is cardiorespiratory by nature, caused by the presence of adults and by the migration of larvae [8,9]. Fatal infection is often associated with severe respiratory compromise, pulmonary hypertension, right heart failure, and/or hemostatic dysfunction with fatal bleeding [10]. However, some infected animals may not show clinical signs during infection, or these infection may be non-specific, making diagnosis difficult [11,12,13].

In Europe, there has been a sharp increase in the number of published cases of carnivores infected by this parasite, so canine angiostrongylosis should be considered an emerging disease [2,14]. It is likely that this increment is influenced by increases in temperature and humidity, due to climate change, which favor the development of intermediate hosts. In addition, other factors may be influencing this emergence, such as anthropogenic factors due to changes in the use and management of land and water, or the increasingly marked urban nature of the fox, a definitive host of the parasite [2,15]. 

In Spain, *A. vasorum* has been described in foxes in the northern half of the Iberian Peninsula and in Murcia, with its prevalence ranging from 1.8% to 43.2% [16,17,18,19,20]. Other studies have also shown *A. vasorum* infections in wolves from the north and north-west (Asturias, Galicia, and Castilla and León) with the prevalence ranging from 2.1% to 11.6% [21,22]. Badgers have also been shown to be a definitive host of this parasite throughout the Mediterranean area [23]. More recently, two studies have been carried out in dogs in various regions of the Iberian Peninsula, where the presence of *A. vasorum* had been found in wild carnivores. These studies were carried out in the northern third of the Iberian Peninsula, as well as some central and coastal Mediterranean regions, reporting variable prevalence ranging from 1% to 2.74% and suggesting that this disease is underestimated in dogs [7,15]. Given the emerging nature of the disease and the results obtained previously in some Spanish regions, the objective of this study was to complete and update an epidemiologic map of *A. vasorum* in dogs in Spain.

## 2. Materials and Methods

### 2.1. Study Area and Climate 

Spain is located in the south of Western Europe and next to North Africa, with an area of 505,370 km², being the fourth largest country in continental Europe, after Russia, Ukraine, and France [24]. Spain is organized into seventeen autonomous communities—with a total of fifty provinces—and two autonomous cities. Fifteen autonomous communities are located on the Iberian Peninsula, while the other two, made up of islands, are located in the western Mediterranean Sea (Balearic Islands) and in the Atlantic Ocean close to the northwest coast of Africa (Canary Islands). The autonomous cities (Ceuta and Melilla) are located in the north coast of the African continent, next to the Mediterranean Sea and bordering Morocco. 

In peninsular Spain, the topographical relief is characterized by being high, with an average altitude of 660 meters above sea level. The mountain systems of Spain are very numerous and occupy almost half of the national territory. The terrain is articulated around a large Meseta Central (central plateau) that occupies most of the center of the Iberian Peninsula. The relief is also characterized by the depression of the Guadalquivir river, located in the southwest of the peninsula, and by the Ebro river depression, in the northeast. Moreover, Spain has a network of rivers and affluents more than 100 km in length, which run through the entire peninsula [25].

According to Köppen’s Climate Classification [26], Spain has a predominantly Mediterranean climate (Cs). Figure 1 shows a map that includes the main climates present in Spain. Most of the central-southern and northeastern coasts of the Iberian Peninsula and the Balearic Islands have a predominantly hot-summer Mediterranean climate (Csa), with mild temperatures in winter, high temperatures in summer, and irregular rainfall, depending on the geographical location. In the western autonomous communities (Asturias, Cantabria, Basque Country, La Rioja, and Navarre; northern and southern Aragon; as well as central Catalonia and the north-east of Castilla and León), the oceanic climate (Cfb) predominates, characterized by abundant rainfall throughout the year—especially in winter—and cool temperatures. The climate that predominates in Galicia and the rest of Castilla and León is a warm-summer Mediterranean climate (Csb), a temperate climate with dry and cool summers. In the southeastern coastal regions, adjoining inland areas, and the center of Aragon, the predominant climate is the cold steppe climate (BSk), a dry climate characterized as such because the amount of evaporation exceeds that of precipitation, on average. The Canary Islands have a desert climate (BWh) along the southern coasts and on the flatter islands. The mountainous islands have a hot semi-arid climate (steppe) (BSh) and BSk in the northern and inland zones, and Csa and Csb climates in the center (mountain tops) of the islands.

### 2.2. Sample Collection and Analysis

From January 2020 to March 2022, a total of 5619 blood samples from dogs from all autonomous communities and autonomous cities of Spain were collected. Samples were collected in 62 veterinary clinics and hospitals, which voluntarily collaborated with the study. Samples were randomly collected, provided they met the inclusion criteria. This meant that samples were obtained from both apparently healthy dogs and dogs with various clinical signs and that they were representative of the canine population admitted to Spanish practices. The distribution of collected samples is described in Table 1 and Table 2. Epidemiological data such as age, sex, and habitat were recorded for each dog. To analyze the results according to age, the dogs were grouped into 5 age groups (<1 year; 1–5 years; >5–10 years; >10–15 years and >15 years). The criteria for inclusion were (a) no previous history of angiostrongylosis, (b) not receiving prophylactic treatment against *A. vasorum* regularly, and (c) owner consent to participate in the survey. 

Blood samples were collected from the cephalic or jugular vein, placed in 3 mL serum tubes, and centrifuged. Serum samples were kept at −20 °C until tests were performed. All samples were tested for the presence of circulating antigens of *A. vasorum* using Angio Detect ^TM^ (IDEXX Laboratories Inc., Westbrook, ME, USA). This qualitative rapid test was carried out following the manufacturer’s instructions. 

For this study, no ethical approval was required, since all blood samples were routinely collected for official diagnostic and monitoring purposes and subsequently made available to this study. All animals were handled according to the principles of animal care required by Spanish Royal Decree 53/2013, and the study was carried out in accordance with the current Spanish and European legislation on animal protection.

### 2.3. Statistical Analysis

Data were analyzed using GraphPad Prism 8.4 (GraphPad, San Diego, CA, USA). Descriptive analysis of the variables considered was carried out considering the proportions of the qualitative variables. Chi-square and Fisher’s exact tests were performed to compare the proportions: Fisher’s exact test was used as a post hoc analysis of the Chi-square test. Confidence interval (95% CI) values were also calculated. In all cases, the significance level was established at *p* < 0.05.

## 3. Results

The overall apparent prevalence of canine angiostrongylosis in Spain was 1.39% (CI 95%:1.11–1.73%). The autonomous communities with the highest prevalence were Murcia (4.12%), Basque Country (3.25%), Asturias (2.50%), and Cantabria (2.40%) (Table 1). The remaining autonomous communities showed prevalence <2%, while in the Balearic Islands and in the autonomous cities of Ceuta and Melilla, no dogs were found to be positive for canine angiostrongylosis. By province, the highest prevalence was found in Bizkaia (4.80%), Murcia (4.12%), Huelva (3.85%), Cádiz (3.45%), and Teruel (3.33%) (Table 2, Figure 2). In the Canary Islands, only one dog was reported to be positive for canine angiostrongylosis, which resided on the island of Tenerife.

Table 3 shows the results based on sex, age, habitat, and climate. No statistically significant differences were found between males and females (x^2^ = 0.1631, df = 1, *p* = 0.6863). The highest prevalence was found in dogs between 1 and 10 years old, although no significant differences were found between age groups (x^2^ = 6.237, df = 4, *p* = 0.1822). Regarding the habitat of the dogs analyzed, significant differences were observed between the different groups (x^2^ = 8.564, df = 2, *p* = 0.0138), with statistically significant differences between indoor/outdoor and outdoor dogs (*p* = 0.0038) but not between indoor and indoor/outdoor dogs (*p* = 0.7693) or between indoor and outdoor dogs (*p* = 0.2355). When the results were analyzed according to the climate, significant differences were observed between the different groups (x^2^ = 11.12, df = 5, *p* = 0.0490), with statistically significant differences between Csa and BSh (*p* = 0.0257), between Csb and BSh (*p* = 0.0306), between Cfb and BSh (*p* = 0.0029), and between BSk and BSh (*p* = 0.0183). The highest prevalence was observed in the Cfb climate.

## 4. Discussion

This study provides a complete epidemiological map of canine angiostrongylosis in Spain, showing the presence of this parasite in a large part of the national territory. There have been several studies carried out in European countries that demonstrated its wide presence in countries such as Austria, Germany, Italy, Slovakia, Switzerland, Hungary, and the United Kingdom, with the prevalence ranging from 0.15% to 1.76%. In these studies, diagnoses were made based on serum antigen detection, and a prevalence similar to those in this study was reported [27,28,29,30,31,32,33], demonstrating that although prevalence is generally low, this disease is widely distributed throughout much of the continent, showing its emerging nature. Despite this, to date, there have only been two previous studies in Spain that have evaluated the presence of this disease in different regions, so an epidemiological map was needed to help determine the current status of canine angiostrongylosis in Spain [7,8,9,10,11,12,13,14,15].

When comparing the current results with reports of canine angiostrongylosis in regions that had already been studied previously, it is observed that the prevalence has remained at similar values, probably because these are relatively recent studies with similar inclusion criteria [7,15]. Previously, epidemiological data in wild carnivores have been published, demonstrating the presence of *A. vasorum* in regions where canine angiostrongylosis has been reported. This is the case for the autonomous communities with a Cfb climate, where previous studies have shown a high presence of this parasite in wolves and foxes [16,21]. The high rainfall that occurs in the north of the country and the extensive presence of vegetation favor the development of intermediate hosts and the infection of wild carnivores [21], which in turn act as definitive hosts favoring the infection of dogs that live in the areas, especially in rural areas.

In regions with a Bsk climate, a high prevalence has also been obtained, especially in Murcia. In these areas, the climate is semi-arid, with little annual rainfall. However, in these regions, there are large irrigated areas, such as the Ebro Valley and Murcia, with various crop protection techniques such as greenhouses, macro-tunnels (large structures made of arches of PVC pipes, usually covered with one or more layers of greenhouse plastic, agrotextile, or insect-proof netting) or micro-tunnels (plastic or spun-bonded fabric sheets placed on metal hoops over developing crops), which create the necessary conditions for the life cycle of gastropod molluscs to develop.

In general, the prevalence obtained in provinces with a Csa climate varied between 1% and 2%, being higher in some provinces, such as Huelva or Cádiz, and lower in other provinces (Madrid), whereas antigen positivity was not found in three provinces only (i.e., Girona and Balearic Islands). Given that only the predominant climates of each province have been considered, ignoring the variability of climates, and the hydrographic and orographic conditions present in each one, it is understandable that there was some variability in the results. For example, Huelva (prevalence 3.85%) and Cádiz (3.45%) have strong hydrographic components, with rivers, natural or artificial stagnant waters, marshes, and national natural parks, which, together with high temperatures, favor the development of the intermediate host.

In the Canary Islands, a case of angiostrongylosis was reported for the first time. Although this region was traditionally considered to be free of this disease, a recent study has shown the presence of the third-stage larvae of *A. vasorum* in different gastropods, such as *Plutonia lamarckii* and *Cornu aspersum*, in Tenerife and El Hierro [34]. 

The diagnostic method used has been shown to have a sensitivity of 98.1% and a specificity of 99.4% [35,36,37], which allows us to carry out a complete, precise, and exhaustive survey of the distribution of *A. vasorum* in Spain, thanks to excellent positive and negative predictive values (96.35% and 99.69%, respectively). The results of this study have been analyzed based on the calculation of the apparent prevalence, which has allowed us to compare and discuss the values obtained from previous studies carried out both in Spain and in other European countries [7,15,27,28,29,30,31,32,33]. However, with this diagnostic method, it is possible that some infected dogs go unnoticed, since studies have shown that experimental infections can display a negative result sooner than 9 weeks post-infection [36]. For this reason, in the case of a clinical suspicion, it is recommended to carry out diagnostic methods considered to be the gold standard, such as the Baermann test and the FLOTAC technique [38].

The results according to age indicate that the highest prevalence was found in dogs between 1 and 10 years of age. Younger dogs (less than 18 months) are generally considered to be at a higher risk of infection [33]. However, the prevalence shown by dogs less than 12 months of age in this study is lower, although in order to analyze these results, the limitations of the diagnostic test used must be taken into account, as mentioned above.

In general, there are few provinces in Spain that have obtained an apparent prevalence of 0%, which indicates how widely distributed this parasite is in the country. Canine angiostrongylosis presents a very varied clinical picture, which can make a clinical suspicion tricky for practitioners not aware of the disease. Both its respiratory and neurological clinical presentation or its presentation as a hemorrhagic diathesis can be easily confused with other more frequent or better-known pathologies. For this reason, it is necessary to create awareness and knowledge of the presence of this parasite among Spanish veterinarians, with epidemiological studies and follow-ups that show the distribution and dynamics of *A. vasorum*. In addition, veterinarians should include this pathology in the differential diagnosis of dogs that present compatible signs, and prevention campaigns should be promoted for animals that are at risk of infection. 

## 5. Conclusions

This study presents a completed distribution map of *A. vasorum* in Spain in dogs. The parasite is heterogeneously present in a large part of the territory, demonstrating its expansion throughout the country. Bearing in mind that it is a fairly unknown disease among clinical veterinarians and among dog owners, it is quite understandable that the prevention of this pathology is not being carried out correctly in most animals, especially in those that are at higher risk. For this reason, it is necessary to carry out campaigns that stimulate preventive actions to avoid infections. Moreover, the realization of a risk map would be extremely useful, taking the bioclimatic variables provided in this study into account. For this, it would be necessary to analyze other variables (bioclimatic, vegetation, surface and underground water masses, habitats of intermediate hosts, land uses, biogeographic regions, etc.), which would expand the information provided in this study. In addition, considering that it is an emerging disease that is spreading throughout Europe, more studies are needed to address the distribution of *A. vasorum* in Spain, both in wild carnivores and in dogs, in order to understand the dynamics of its evolution.

## Figures and Tables

**Figure 1 animals-12-02217-f001:**
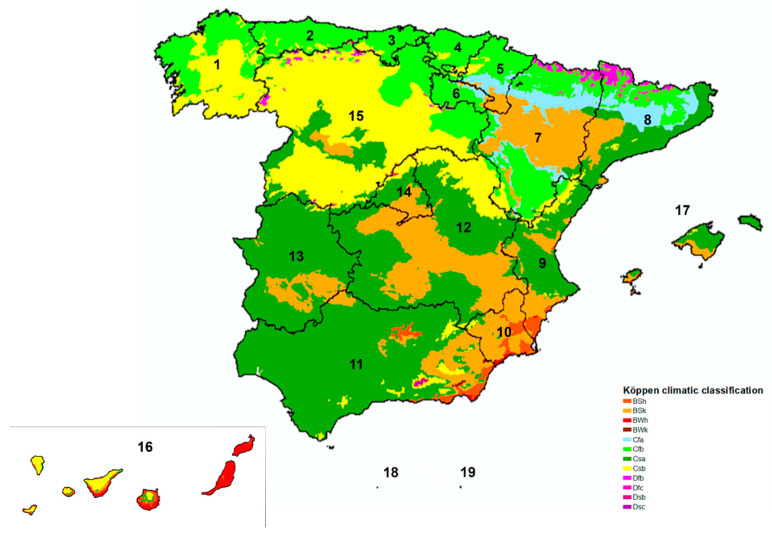
Map with the main climates present in Spain (Köppen Climate Classification System). The numbers correspond to the autonomous communities and autonomous cities, as described in Table 1. *Abbreviations*: BSh: hot semi-arid climate (steppe); BSk: cold steppe climate; BWh: desert climate; BWk: Cold desert climate; Cfa: humid subtropical climate; Cfb: oceanic climate; Csa: hot-summer Mediterranean climate; Csb: warm-summer Mediterranean climate; Dfb: humid continental climate; Dfc: subarctic climate; Dsb: humid continental climate; Dsc: subarctic climate.

**Figure 2 animals-12-02217-f002:**
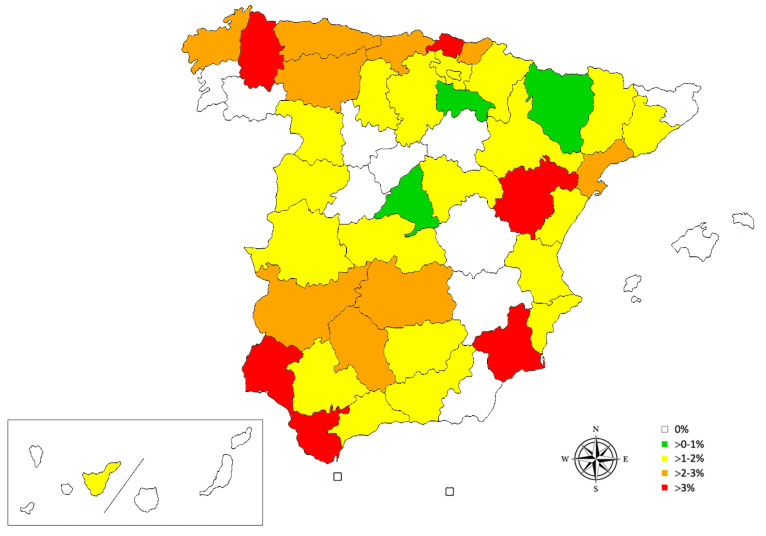
Prevalence map for *A. vasorum* in domestic dogs in Spain by province and the autonomous cities of Ceuta and Melilla.

**Table 1 animals-12-02217-t001:** Prevalence of *A. vasorum* in dogs in Spain by autonomous communities and autonomous cities (Ceuta and Melilla). *Abbreviations*: n, number of dogs sampled; +, number of positive dogs; %, percentage of positive dogs (confidence interval 95%).

	n	+	% (CI 95%)
** 1. Galicia**	428	8	1.87 (0.95–3.64)
** 2. Asturias**	120	3	2.50 (0.85–7.09)
** 3. Cantabria**	125	3	2.40 (0.82–6.82)
** 4. Basque Country**	308	10	3.25 (1.77–5.87)
** 5. Navarre**	170	3	1.76 (0.60–5.06)
** 6. La Rioja**	100	1	1.00 (0.18–5.45)
** 7. Aragón**	209	3	1.44 (0.49–4.13)
** 8. Catalonia**	521	6	1.15 (0.53–2.49)
** 9. Valencian Community**	352	5	1.42 (0.61–3.28)
** 10. Murcia**	97	4	4.12 (1.62–10.13)
** 11. Andalusia**	502	10	1.99 (1.09–3.63)
** 12. Castilla-La Mancha**	304	4	1.32 (0.51–3.33)
** 13. Extremadura**	153	3	1.96 (0.67–5.61)
** 14. Madrid**	389	3	0.77 (0.26–2.24)
** 15. Castilla and León**	985	11	1.12 (0.62–1.99)
** 16. Canary Islands**	552	1	0.18 (0.03–1.02)
** 17. Balearic Islands**	229	0	0.00 (0.00–1.65)
** 18. Ceuta**	27	0	0.00 (0.00–12.45)
** 19. Melilla**	48	0	0.00 (0.00–7.41)
** Total**	**5619**	**78**	**1.39 (1.11–1,73)**

**Table 2 animals-12-02217-t002:** Prevalence of *A. vasorum* in dogs in Spain by province and autonomous cities (Ceuta and Melilla). *Abbreviations*: n, number of dogs sampled; +, number of positive dogs; %, percentage of positive dogs (confidence interval 95%).

Autonomous Community Province	n	+	% (CI 95%)	Autonomous Community Province	n	+	% (CI 95%)
**Galicia**				**Extremadura**			
A Coruña	224	5	2.23 (0.96–5.12)	Cáceres	71	1	1.41 (0.25–7.56)
Lugo	98	3	3.06 (1.05–8.62)	Badajoz	82	2	2.44 (0.67–8.46)
Ourense	47	0	0.00 (0.00–7.56)	**Andalusia**			
Pontevedra	59	0	0.00 (0.00–6.11)	Cádiz	58	2	3.45 (0.95–11.73)
**Asturias**				Seville	70	1	1.43 (0.25–7.66)
Oviedo	120	3	2.50 (0.85–7.09)	Málaga	54	1	1.85 (0.33–9.77)
**Cantabria**				Granada	87	1	1.15 (0.20–6.23)
Santander	125	3	2.40 (0.82–6.82)	Jaén	67	1	1.49 (0.26–7.98)
Basque Country				Huelva	78	3	3.85 (1.32–10.71)
Araba	81	1	1.23 (0.22–6.67)	Almería	41	0	0.00 (0.00–8.57)
Bizkaia	125	6	4.80 (2.22–10.08)	Córdoba	47	1	2.13 (0.38–11.11)
Gipuzkoa	102	3	2.94 (1.01–8.29)	**Castilla-La Mancha**			
**Navarre**				Albacete	40	0	0.00 (0.00–8.76)
Navarre	170	3	1.76 (0.60–5.06)	Guadalajara	87	1	1.15 (0.20–6.23)
**La Rioja**				Cuenca	38	0	0.00 (0.00–9.18)
La Rioja	100	1	1.00 (0.18–5.45)	Ciudad Real	85	2	2.35 (0.65–8.18)
Aragón				Toledo	54	1	1.85 (0.33–9.77)
Huesca	112	1	0.89 (0.16–4.88)	**Castilla y León**			
Zaragoza	67	1	1.49 (0.26–7.98)	León	148	3	2.03 (0.69–5.79)
Teruel	30	1	3.33 (0.59–16.67)	Zamora	94	1	1.06 (0.19–5.78)
**Catalonia**				Salamanca	137	2	1.46 (0.40–5.17)
Girona	75	0	0.00 (0.00–4.87)	Valladolid	85	0	0.00 (0.00–4.32)
Barcelona	285	3	1.05 (0.36–3.05)	Palencia	112	2	1.79 (0.49–6.28)
Tarragona	91	2	2.20 (0.60–7.66)	Burgos	154	3	1.95 (0.66–5.57)
Lleida	70	1	1.43 (0.25–7.66)	Segovia	98	0	0.00 (0.00–3.77)
**Valencian Community**				Soria	82	0	0.00 (0.00–4.47)
Castellón	60	1	1.67 (0.29–8.85)	Ávila	75	0	0.00 (0.00–4.87)
Valencia	207	3	1.45 (0.49–4.17)	**Canary Islands**			
Alicante	85	1	1.18 (0.21–6.37)	Las Palmas	240	0	0.00 (0.00–1.58)
**Murcia**				S/C Tenerife	312	1	0.32 (0.06–1.79)
Murcia	97	4	4.12 (1.62–10.13)	**Balearic Islands**			
**Madrid**				Balearic Islands	229	0	0.00 (0.00–1.65)
Madrid	389	3	0.77 (0.26–2.24)	**Ceuta**	27	0	0.00 (0.00–12.45)
				**Melilla**	48	0	0.00 (0.00–7.41)
				**Total**	**5619**	**78**	**1.39**

**Table 3 animals-12-02217-t003:** Prevalence of A. vasorum in dogs in Spain by sex, age, habitat, and climate. Abbreviations: n, number of dogs sampled; +, number of positive dogs; %, percentage of positive dogs (confidence interval 95%); BSh: hot semi-arid climate (steppe); BSk: cold steppe climate; BWh: desert climate; BWk: Cold desert climate; Cfb: oceanic climate; Csa: hot-summer Mediterranean climate; Csb: warm-summer Mediterranean climate.

	n	+	% (CI 95%)
**Sex**			
Female	2826	41	1.45 (1.07–1.96)
Male	2793	37	1.32 (0.96–1.82)
**Age**			
<1 year	915	7	0.77 (0.37–1.57)
1–4.9 years	1183	21	1.78 (1.16–2.70)
5–9.9 years	1389	24	1.73 (1.16–2.56)
10–15 years	1206	17	1.41 (0.88–2.25)
>15 years	926	9	0.97 (0.51–1.84)
**Habitat**			
Outdoor	3327	59	1.77 (1.38–2.28)
Indoor	431	4	0.93 (0.36–2.36)
Indoor/Outdoor	1861	15	0.81 (0.49–1.33)
**Climates**			
Csa	2242	29	1.29 (0.90–1.85)
Csb	1197	16	1.34 (0.82–2.26)
Cfb	1271	26	2.05 (1.40–2.98)
BSk	417	7	1.68 (0.82–3.42)
BWh	143	0	0.00 (0.00–2.62)
BSh	349	0	0.00 (0.00–1.09)

## Data Availability

The data presented in this study are available from the corresponding author upon request.

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
