# Peer review of "Comprehensive Map of Canine Angiostrongylosis in Dogs in Spain"

_animals, 2022, doi:10.3390/ani12172217_

Round 1

Reviewer 1 Report

Animals-1832066

Comprehensive map of canine angiostrongylosis in dogs in Spain

The great strength of this study is the large sample size. However, the results only confirm the prevalence previously reported in Spain and in many parts of Europe. In addition, the same authors report the first epidemiological study in Spain of A. vasorum in 2020 [Carretón, et al (2020) First epidemiological survey of Angiostrongylus vasorum in domestic dogs from Spain. Parasites & Vectors. 202013, 306] with very similar results to those now reported. Then in 2021 part of the same authors report another epidemiological study in some regions of Spain [Morchón, et al (2021) Angiostrongylus vasorum in domestic dogs in Castilla y León, Iberian  Peninsula, Spain. Animals. 11(6), 1513], with very similar results to those now reported. Therefore, this new study does not provide more findings than those previously reported.

Some specific comments:

Materials and methods

Line 127. Inclusion criteria. Collecting samples from clinics can produce a bias because the dogs are brought in for some clinical condition, therefore the selection is not random.

Line 137. In my opinion, an informed consent should have been signed by the owners to authorize the use of their dog's samples for other purposes; informed consent is part of ethical approval.

Line 146. Significance level at p<0.005??? this must be a mistake.

Results

I do not consider that the results have to be reported by autonomous communities and then subdivided by autonomous cities; the parasites in this case are not distributed by political divisions but by environmental conditions that can include various communities. Considering the above, perhaps only Table 1 should be included and a map with the prevalence according to climatic conditions could be considered, as reported in Table 3. However, this has already been reported in Morchon et al, 2021.

In all the tables and figures, the title indicates the prevalence of D. immitis. In Table 3, the level of significance should have been indicated together with the odd ratio and confidence interval.

Discussion

The study confirms what was previously published in Spain and does not provide new findings that complement the epidemiological map. It is not very relevant to discuss the differences between 1% and 2% or very similar prevalence.

Author Response

All changes suggested by the reviewer have been added to the text and marked in red. In addition, the article has been reviewed by a proofreading /editing service. The authors thank the referee for his dedication and time spent reviewing this article.

The great strength of this study is the large sample size. However, the results only confirm the prevalence previously reported in Spain and in many parts of Europe. In addition, the same authors report the first epidemiological study in Spain of A. vasorum in 2020 [Carretón, et al (2020) First epidemiological survey of Angiostrongylus vasorum in domestic dogs from Spain. Parasites & Vectors. 202013, 306] with very similar results to those now reported. Then in 2021 part of the same authors report another epidemiological study in some regions of Spain [Morchón, et al (2021) Angiostrongylus vasorum in domestic dogs in Castilla y León, Iberian  Peninsula, Spain. Animals. 11(6), 1513], with very similar results to those now reported. Therefore, this new study does not provide more findings than those previously reported.

We consider that this is an article that provides valuable and novel information. The studies cited show results carried out in specific points of the Spanish geography. However, this study completes the epidemiological map of A. vasorum in Spain. This is an epidemiological map that has never been carried out in Spain, and that has been carried out in very few European countries, which justifies the need for more studies on this parasite. Moreover, the number of samples constitutes one of the highest in epidemiological studies carried out worldwide.

A. vasorum has been a neglected parasite in Spain for many years, for various reasons, as detailed in the paper, but undoubtedly one of them has been the lack of epidemiological studies that demonstrate the presence of this parasite to clinicians. We believe that this is a pioneering study, which will be very useful not only scientifically, but clinically so that veterinarians will consider angiostrongylosis when they have a canine patient with compatible symptoms. Furthermore, since it is an emerging disease in Europe, as other authors have pointed out before, we consider that these epidemiological data are of great scientific value.

Some specific comments:

Materials and methods

Line 127. Inclusion criteria. Collecting samples from clinics can produce a bias because the dogs are brought in for some clinical condition, therefore the selection is not random.

Surely the sampling is not exactly representative of the dog population of the country, but we consider that it is very close. The samples collected in veterinary clinics serve various profiles, not only animals that come to the clinic because they are sick. For example, we also included samples of dogs that were collected by animal shelters which collaborate with different veterinary clinics to sanitize the dogs that they collect. The animal shelters/organizations in Spain have an important weight, there are many associations that are created with the sole objective of collecting animals from the street and from shelters, to sanitize them and promote their national and international adoption. The levels of collaboration of these associations with veterinary clinics is logically very high, which is why these animals are also represented in the study. In addition, in almost all the autonomous communities of Spain, the rabies vaccine is compulsory, under penalty of a fine. This forces dog owners to take the dog at least once a year to vaccinate, even those who have no interest in the welfare of their pets. Thus, these kinds of owners, who do not keep their dog properly vaccinated or dewormed, are also represented by this study. Of course, healthy animals from responsible owners who come to appointments oriented towards preventive medicine were also included.

Line 137. In my opinion, an informed consent should have been signed by the owners to authorize the use of their dog's samples for other purposes; informed consent is part of ethical approval.

As indicated in the Ethics statements, all the owners were informed and signed a document to this effect, as it appears in our manuscript. In addition, the journal has the model of the document of consent. This study has scrupulously followed current Spanish and European legislation regarding ethics and animal welfare.

Line 146. Significance level at p<0.005??? this must be a mistake.

True, the correct value would be p < 0.05 and has been modified in the text.

Results

I do not consider that the results have to be reported by autonomous communities and then subdivided by autonomous cities; the parasites in this case are not distributed by political divisions but by environmental conditions that can include various communities. Considering the above, perhaps only Table 1 should be included and a map with the prevalence according to climatic conditions could be considered, as reported in Table 3. However, this has already been reported in Morchon et al, 2021.

Logically, parasites know nothing about borders, which is why we have described the results obtained based on the different main climates in Spain and we have discussed the results based on this. But at an informative and clarifying level, we consider it important to also present the results according to each autonomous community and province, since this will allow clinical veterinarians and health authorities to interpret and disseminate the results.  

In all the tables and figures, the title indicates the prevalence of D. immitis. In Table 3, the level of significance should have been indicated together with the odd ratio and confidence interval.

Thanks for your observation. We have fixed this error.

Discussion

The study confirms what was previously published in Spain and does not provide new findings that complement the epidemiological map. It is not very relevant to discuss the differences between 1% and 2% or very similar prevalence.

As we have commented previously, we have not limited these results to discussing the prevalence values ​​obtained based on previous studies, but also the findings obtained in those regions in which this disease had never been studied. We consider this important, especially given the emerging nature of this disease.

Reviewer 2 Report

The paper “Comprehensive map of canine angiostrongylosis in dogs in Spain” by Elena Carreton and co-authors is a nice piece of work that fills the gap of angiostrongylosis in Spain. I believe this paper entails a great amount of effort and patience with the veterinary practitioners but the result is a nice picture of the disease in this country. However, some improvements could be done to make the best of this data.

Minor issues

1.       English needs improvement, I recommend passing the paper to a native colleague.

2.       The use of the word “reservoir” (lines 67, 72, 203, 220), is it appropriate considering the life cycle of this parasite? Would definitive host be better?

3.       Please include in the Ethical statement a sentence regarding why the Ethical approval was waived (number of the Spanish/European law), plus the sentence in the main text.

4.       Name of the communities/regions is not consistent in my opinion, for example Basque Country and Navarra (shouldn’t it be Navarre?), Sevilla (Seville?) etc. Please correct the names to make them all Spanish or English/International equivalent. Also some region names are cincomplete (Region of Murcia? Community of Madrid?)

5.       Line 230, test usually  yields false negatives before 9 months post infection, so it’s not surprising that prevalence in dogs under 1 year is lower than expected, as you say a few lined below.

6.       Dog breed was mentioned as one of the studied parameters but it was not mentioned anymore in the text.

Major issues

1.       The AngioDetect test interpretation should be briefly described in the paper (cutoff values?)

2.       Although the maps included in the text are useful, I believe it would be much better to make a risk map of angiostrongyliasis in the paper that includes the relevant variables (ie seroprevalence, bioclimatic features…) since the province-colour map and the bioclimatic maps are very limited on their own.

Author Response

Thank you for your comment. All changes suggested have been added to the text and marked in red. In addition, the article has been reviewed by a proofreading /editing service. The authors thank the referee for his dedication and time spent reviewing this article.

The paper “Comprehensive map of canine angiostrongylosis in dogs in Spain” by Elena Carreton and co-authors is a nice piece of work that fills the gap of angiostrongylosis in Spain. I believe this paper entails a great amount of effort and patience with the veterinary practitioners but the result is a nice picture of the disease in this country. However, some improvements could be done to make the best of this data.

Minor issues

English needs improvement, I recommend passing the paper to a native colleague.

We have checked the paper with a professional editor and fixed some grammar errors

The use of the word “reservoir” (lines 67, 72, 203, 220), is it appropriate considering the life cycle of this parasite? Would definitive host be better?

Thank you for your comment. We have replaced “reservoir” by “definitive host” an “reservoirs of water” by “natural or artificial stagnant waters”

Please include in the Ethical statement a sentence regarding why the Ethical approval was waived (number of the Spanish/European law), plus the sentence in the main text.

In Spain, Royal Decree 53/2013 regulates and define animal experimentation. This has been added to the text.

Name of the communities/regions is not consistent in my opinion, for example Basque Country and Navarra (shouldn’t it be Navarre?), Sevilla (Seville?) etc. Please correct the names to make them all Spanish or English/International equivalent. Also some region names are cincomplete (Region of Murcia? Community of Madrid?)

Minor errors in Seville and Navarre have been fixed. The rest of the names are the accepted ones. The Region of Murcia is also the Autonomous Community of Murcia and the Community of Madrid and the Autonomous Community of Madrid, but from a general/popular point of view they are called as described in the article.

Line 230, test usually  yields false negatives before 9 months post infection, so it’s not surprising that prevalence in dogs under 1 year is lower than expected, as you say a few lined below.

The reviewer is correct, and indeed this has been taken into account when discussing the results by age.

Dog breed was mentioned as one of the studied parameters but it was not mentioned anymore in the text.

It's true. We have eliminated this variable from materials and methods since it has not been analyzed in the manuscript at any time. 

Major issues

The AngioDetect test interpretation should be briefly described in the paper (cutoff values?)

We have added “Three drops of serum and three drops of supplied buffer were added to each test and waited for 5 minutes. When two bands were observed in the test, the sample was reported as positive, but when only one band was observed in the control zone, it was reported as negative. All other results, if any, invalidated the sample result”. As mentioned in this description, there are no cutoff values ​​as it is a semi-quantitative test.

Although the maps included in the text are useful, I believe it would be much better to make a risk map of angiostrongyliasis in the paper that includes the relevant variables (ie seroprevalence, bioclimatic features…) since the province-colour map and the bioclimatic maps are very limited on their own.

This is an excellent suggestion and indeed we are considering it for future studies, but it is not the aim of this study nonetheless. The aim was to describe the prevalence obtained based on some variables, but it was not intended to determine risk maps. To do this we would need a series of data (bioclimatic data obtained through the geographic information system (GIS), exact geolocation of the positive dogs, etc.) that we do not have for this study. We are aware that this has some limitations, but we consider that the information provided in this study is of great value per se, since the epidemiology of angiostrongylosis in Spain had never before been reported at a national level.

Reviewer 3 Report

This manuscript describes an evaluation of canine angiostrongylosis in dogs in Spain.  This is well written and worthy of publication. There are, however, a few items that need to be addressed, which are described below.

Line 17. It would be better if the authors define the time period as “January 2020 through March 2022”, rather than “last two years”.

Lines 19 and 29 list two different prevalence numbers.

Lines 144-148: I assume the authors are using Fisher’s Exact test as a post-hoc analysis of chi-square.  If so, this should be spelled out.  Also, the authors calculate an apparent prevalence and should describe that here as well.  It would be better to also calculate the true prevalence by using TP = (AP+Sp-1)/(Se+Sp-1).  Since the authors mention the test’s sensitivity and specificity (lines 226-7), the overall total prevalence of canine angiostronylosis in dogs in Spain is 0.81%. (Blaker's, Sterne, Clopper-Pearson and Wilson confidence limits are calculated as described by Reiczigel, Földi, Ózsvári (2010). Exact confidence limits for prevalence of a disease with an imperfect diagnostic test, Epidemiology and Infection 138:1674-1678.)

Line 148. I assume the authors intended on setting significance at P < 0.05 and not P < 0.005.

Lines 150-181. Please correct all prevalence data to total prevalence as all of these numbers are apparent prevalence data.

Tables 1-3 and Figure 2 mention D. immitis and not A. vasorum. Please correct.

Lines 168-169. Please provide the X2, df, and P for sex

Lines 169-171. Please provide the X2, df, and P for age

Line 175. What are the P values of indoor vs indoor/outdoor and indoor vs outdoor? This may be accomplished easiest by adding another column in Table 3 for the P values.

Author Response

Thank you for your comments. All changes suggested have been taken into account. In addition, the article has been reviewed by a proofreading /editing service. The authors thank the referee for his dedication and time spent reviewing this article.

This manuscript describes an evaluation of canine angiostrongylosis in dogs in Spain.  This is well written and worthy of publication. There are, however, a few items that need to be addressed, which are described below.

Line 17. It would be better if the authors define the time period as “January 2020 through March 2022”, rather than “last two years”.

Corrected, thank you

Lines 19 and 29 list two different prevalence numbers.

Big mistake, thanks for noticing.

Lines 144-148: I assume the authors are using Fisher’s Exact test as a post-hoc analysis of chi-square.  If so, this should be spelled out.  

Corrected in the text

Also, the authors calculate an apparent prevalence and should describe that here as well.  It would be better to also calculate the true prevalence by using TP = (AP+Sp-1)/(Se+Sp-1).  Since the authors mention the test’s sensitivity and specificity (lines 226-7), the overall total prevalence of canine angiostronylosis in dogs in Spain is 0.81%. (Blaker's, Sterne, Clopper-Pearson and Wilson confidence limits are calculated as described by Reiczigel, Földi, Ózsvári (2010). Exact confidence limits for prevalence of a disease with an imperfect diagnostic test, Epidemiology and Infection 138:1674-1678.)

We understand the reviewer's reasoning, but prefer to keep the apparent prevalence data. This is so because all the epidemiological studies carried out in the study of this parasite have been carried out in this way. Thus, apparent prevalence data allows us to compare and discuss the values ​​obtained based on previous studies, carried out both in Spain and in other European countries. To this aim, we modified the text making it clear that the results correspond to apparent prevalence.

Line 148. I assume the authors intended on setting significance at P < 0.05 and not P < 0.005.

Yes, it has been corrected

Lines 150-181. Please correct all prevalence data to total prevalence as all of these numbers are apparent prevalence data.

As we have previously commented, we prefer to work with these apparent prevalence data, if the reviewer accepts our reasoning.

Tables 1-3 and Figure 2 mention D. immitis and not A. vasorum. Please correct.

Clumsy mistake, thank you

Lines 168-169. Please provide the X2, df, and P for sex

Provided as (χ2 =0.1631, df=1, p=0.6863)

Lines 169-171. Please provide the X2, df, and P for age

Provided as (χ2 = 6.237, df=4, p=0.1822).

Line 175. What are the P values of indoor vs indoor/outdoor and indoor vs outdoor? This may be accomplished easiest by adding another column in Table 3 for the P values.

We have introduced in the manuscript: “…not between indoor and indoor/outdoor dogs (p=0.7693) or between indoor and outdoor dogs (p=0.2355)”.

Round 2

Reviewer 1 Report

Thank you to the authors for their comments. However, I still  do not see any further improvement on the manuscript. I still believe that with the previously published information is enough to have an epidemiological panorama of the spread of this nematode in Spain. I also believe that the way of sampling is very heterogeneous and can be a cause of bias.

Author Response

As we have previously commented to this reviewer, we try again to clarify or point out the items that concern the reviewer:

  1. “I still believe that with the previously published information is enough to have an epidemiological panorama of the spread of this nematode in Spain”: This study includes NEVER analyzed data from half of Spain. We are talking about a country of more than 500,000 km²: Andalusia, Castilla la Mancha, Extremadura, part of Aragon, the Balearic Islands, the Canary Islands, Ceuta and Melilla had never been analyzed. We insist on this point: this borne parasite vector had NEVER been studied in Spain until recently and the complete epidemiological situation of this parasite in Spain had NEVER been known until now. From here, in subsequent studies, it will be possible to discuss with scientific knowledge whether or not this parasite is spreading in Spain. Until now, it was not possible to speak of possible (probable) expansion without prior objective and published scientific data. If the reviewer considers that ignoring the epidemiological situation of half the country is justifiable so that its publication is not necessary, we would like a justified explanation to try to understand this reasoning.

  1. I also believe that the way of sampling is very heterogeneous and can be a cause of bias”: again we insist: this study includes dogs that are cared for by their owners, dogs whose owners comply with the most basic care of their dogs, abandoned dogs whose care has been null (many of them without the mandatory identification by microchip or vaccinations), healthy dogs in preventive medicine, dogs with some pathology that caused a visit to the veterinarian, dogs whose owners were just complying with the legal regulations of the country in keeping animals, totally abandoned dogs and "outside the legal circuit" collected by an animal shelter; rural dogs, city dogs, dogs that live in homes and dogs that have lived on the streets until they were taken in by animal shelters. In any of these cases, the way of sampling has been the same: random selection by the collaborating clinics, provided that they meet the inclusion criteria indicated in the article. If the reviewer still considers that the sampling has not been adequate, we will ask them to indicate what sampling method they consider adequate to carry out a study of these characteristics.